# Anti-*Candida* and Anti-Inflammatory Properties of a Vaginal Gel Formulation: Novel Data Concerning Vaginal Infection and Dysbiosis

**DOI:** 10.3390/microorganisms11061551

**Published:** 2023-06-10

**Authors:** Luca Spaggiari, Gianfranco B. Squartini Ramos, Caterina A. Squartini Ramos, Andrea Ardizzoni, Natalia Pedretti, Elisabetta Blasi, Francesco De Seta, Eva Pericolini

**Affiliations:** 1Clinical and Experimental Medicine Ph.D. Program, University of Modena and Reggio Emilia, 41125 Modena, Italy; luca.spaggiari@unimore.it; 2Department of Surgical, Medical, Dental and Morphological Sciences with Interest in Transplant, Oncological and Regenerative Medicine, University of Modena and Reggio Emilia, 41125 Modena, Italy; 238919@studenti.unimore.it (G.B.S.R.); 224739@studenti.unimore.it (C.A.S.R.); andrea.ardizzoni@unimore.it (A.A.); natalia.pedretti@unimore.it (N.P.); elisabetta.blasi@unimore.it (E.B.); 3Department of Medical Sciences, University of Trieste, 34149 Trieste, Italy; 4Institute for Maternal and Child Health, Istituto di Ricovero e Cura a Carattere Scientifico (IRCCS), Burlo Garofolo, 34127 Trieste, Italy

**Keywords:** vaginal infection, *C. albicans*, dysbiosis, topic treatments, anti-inflammatory effect, antifungal effect, vaginal ecosystem, IL-8, vaginal epithelial cells, dimorphic transition, farnesol

## Abstract

Vaginal ecosystem is a unique environment where, in physiological conditions, lactobacilli dominate. However, pathogenic microbial species responsible for vaginitis and vaginosis can also harbor vaginal microbiota. To extend our previously published data, we analyzed here both the anti-*Candida* and anti-inflammatory properties of the vaginal gel formulation, Respecta^®^ Balance Gel (RBG), commercialized as an adjuvant to treat vaginitis and vaginosis. We evaluated its activity by an in vitro model where a monolayer of A-431 vaginal epithelial cells was infected by *Candida albicans* in the presence of RBG or the placebo formulation (pRBG). Specifically, we tested the RBG capacity to counteract *C. albicans* virulence factors and their anti-inflammatory properties. Our results show that, unlike the placebo, RBG reduces *C. albicans* adhesion, its capacity to form hyphae and *C. albicans*-induced vaginal cell damage. Interestingly, both RBG and pRBG reduce LPS-induced IL-8 secretion (with RBG being the most effective), demonstrating that also the placebo retains anti-inflammatory properties. From our experimental approach, we highlighted the possible role of farnesol on such effects, but we would like to point out that lactic acid, polydextrose and glycogen too must be relevant in the actual application. In summary, our results show that RBG impairs *C. albicans* virulence and is able to reduce the inflammation in the vaginal environment, ultimately allowing the establishment of a balanced vaginal ecosystem.

## 1. Introduction

Vulvovaginal candidiasis (VVC) is a painful condition affecting up to 75% of women during their child-bearing age at least once in their lifetime [1]. Furthermore, 5–8% of them develop the chronic form of the disease (RVVC), namely four or more symptomatic episodes per year [2]. A global study published in 2018 reports the worldwide prevalence of RVVC at approximately 138 million women annually, identifying the substantial morbidity and economic burden of the pathology [3]. Vaginal pain, burning sensation, itching and abnormal vaginal discharge are the most typical VVC symptoms, and chronicity to RVVC leads to negative consequences for physical and mental health [4]. VVC onset is due to the pathological overgrowth of *Candida* species, and in 75–90% of cases, *C. albicans* is the species responsible for the disease [5]. This fungus normally dwells in the vaginal mucosa as a commensal, together with the bacterial community, where it establishes a delicate balance supervised by the immune system [6]. *Candida* plays a dual role in the vaginal ecosystem: it can behave as the classic opportunistic pathogen under host dysbiosis conditions, but it can also play a leading role in triggering the VVC in immunocompetent healthy women [7]. What induces *C. albicans* to switch from a harmless commensal to a virulent pathogen during the VVC onset is still unclear. It is widely demonstrated that the disease onset can be facilitated by several predisposing host factors, such as gene polymorphisms, diabetes, prolonged antibiotic therapies, oral estrogen administration, hormone replacement therapy and psychosocial stresses [8,9,10]. As it is a dimorphic fungus, *C. albicans* can assume two different morphologies: yeast cell and hyphal form. Yeast cells are typical of the beneficial *Candida* colonization of the vaginal mucosa. Differently, hyphae produced by yeasts germination are common in symptomatic vaginitis [11,12]. Colonization starts when *Candida* cells adhere to the vaginal mucosa. Dysbiosis conditions, the presence of predisposing factors, or the occurrence of other (yet unknown) events induce the yeasts-to-hyphae transition. Hyphae, which are considered the most adherent morphology of *C. albicans* [13], promote the invasion of the vaginal epithelium [7]. Tissue damage related to the hyphal form is also associated with the production of candidalysin, the only *C. albicans* toxin known to date [14], which is produced by adherent hyphae during epithelial penetration [15]. Candidalysin exerts lytic activity and induces neutrophil recruitment [16] due to proinflammatory cytokines released by damaged vaginal epithelium [17]. In addition, adhesion capacity, extracellular enzymes secretion (such as aspartyl proteases and phospholipase B), biofilm production and several quorum sensing (QS) molecules secreted by *C. albicans* can be considered as other important virulence factors [18]. To date, only four different QS molecules have been identified in the *Fungi* kingdom: farnesol, tyrosol, phenylethanol and tryptophol. These molecules are all involved in inter-microbial communication and regulate *Candida* morphogenesis, virulence and apoptosis [19]. Farnesol is a sesquiterpene alcohol synthesized by *C. albicans* starting from farnesyl pyrophosphate (FPP) [20], and it was the first identified eukaryotic quorum sensing (QS) molecule. Farnesol plays a key role in *C. albicans* virulence by regulating biofilm formation, yeast-to-hyphae transition [21,22] and by affecting the expression of genes involved in the protection of the fungus against oxidative stress [23]. In addition, farnesol may impact the host immune system by affecting macrophages’ phagocytic and antimicrobial potential [24]. Furthermore, its capacity to downregulate the expression of several inflammatory mediators demonstrates that farnesol also has anti-inflammatory activity [25]. The wide spectrum of farnesol biological functions that can be beneficial both for *Candida* and for the host opens to investigate if its employment in antifungal therapy may be beneficial. Despite the fact that *Candida* virulence factors are essential in the early phases of VVC, the chronicity of the disease is widely mediated by the host’s immune response [26]. Specifically, *C. albicans* overgrowth stimulates vaginal epithelial cells to produce alarmins and proinflammatory cytokines, which in turn lead to neutrophil recruitment and consequently to inflammatory symptoms [27]. It is also essential to consider that resident microbiota, often dominated by *Lactobacillus* species [28], could also play a role in the VVC onset. Indeed, numerous studies have demonstrated the antifungal activity of *Lactobacillus* species against *C. albicans* [29,30,31,32] and *non-albicans Candida* species [33,34,35,36]. Lactobacilli may inhibit *Candida* overgrowth by several mechanisms, including the production of lactic acid, which causes the acidification of the vaginal environment. The latter, indeed, is typically acid (pH near 4.0) during almost the entire menstrual cycle. The acidification inhibits *C. albicans* hyphal growth [37,38], which demonstrates the importance of producing acids (such as lactic acid) by resident bacteria in counteracting fungal virulence. Nonetheless, the potential role of the healthy vaginal microbiota in the VVC onset has not yet been completely identified. Respecta^®^ Balance Gel (RBG) is a proprietary topical gel for vaginal health produced by BioFarma S.p.A. (Udine, Italy) and commercialized as an adjuvant for the treatment of vaginitis and vaginosis. The Manufacturers claim that RBG’s healthful effects are due to the restoration of the physiological pH of the vaginal environment, the increased growth of beneficial microorganisms, the hydration and protection of the vaginal tissue and the formation of a protective film. The active components of the gel formulation are polydextrose, lactic acid, farnesol and glycogen. Farnesol activity has been described above. Polydextrose and glycogen act both as prebiotics since they promote eubiosis [39,40]. Lactic acid maintains an acidic vaginal environment in order to counteract possible pathogens [7].

In this work, we analyzed in vitro the antifungal and anti-inflammatory effects of RBG.

## 2. Materials and Methods

### 2.1. Fungal Strains and Growth Conditions

The reference strain *C. albicans* SC5314 (ATCC MYA-2876) and the bioluminescent strain of *C. albicans* gLUC59 (BLI-Ca) were employed in the experimental procedures. Both strains were stored in frozen stocks at −80 °C in Sabouraud Dextrose Broth (Condalab, Madrid, Spain) supplemented with 15% glycerol. After thawing, the fungi were grown in a liquid YPD medium (Yeast extract–Peptose–Dextrose, Scharlab S.L., Barcelona, Spain) and incubated at 37 °C under aerobic conditions for 24 h. Fungi in the exponential growth phase were used in each experiment. *C. albicans* gLUC59 was employed in experimental procedures related to Section 2.5, Section 2.6, Section 2.8 and Section 2.10. *C. albicans* SC5314 was used in experiments described in Section 2.7 and Section 2.9.

### 2.2. Vaginal Epithelial Cells

The A-431 cell line from vaginal epithelial squamous cell carcinoma (ATCC CLR-1555) was employed. These cells were cultured in DMEM (Dulbecco’s Modified Eagle Medium, PAN Biotech, Aidenbach, Germany) supplemented with L-glutamine (2 nM) (Euroclone SpA, Pero, Italy), penicillin (100 U/mL) (Euroclone SpA, Italy), streptomycin (100 μL/mL) (Euroclone SpA, Italy), ciprofloxacin (20 mg/mL) (Euroclone SpA, Italy) and FBS (Fetal Bovine Serum, 10% or 5%, SIGMA-Aldrich, Saint Louis, MO, USA). The cell line was kept viable by subculturing twice a week and incubated at 37 °C with 5% CO_2_.

### 2.3. Respecta^®^ Balance Gel (RBG)

RBG was provided by BioFarma S.p.A. (Udine, Italy) and stored at +4 °C. The Company provided information concerning the components of the gel, even though exact quantities of each component are proprietary information and cannot be reported. In our experiments, 2 different RBG formulations were assessed: in addition to the standard formulation (RBG), a placebo formulation (pRBG) was provided. Compared to RBG, the pRBG lacks polydextrose, glycogen, farnesol and lactic acid. The composition of each RBG formulation is detailed in Table 1. Both RBG and pRBG were diluted in 50 mL of DMEM + 5% of FBS or YPD broth + 10% FBS or Roswell Park Institute da Moore (RPMI) + MOPS (35 mg/mL) + glucose (18 mg/mL), and their final pH was measured by a pH meter (Hanna Instruments, Italy). Dilutions were stored at +4 °C for not more than 2 weeks.

### 2.4. Establishment of A-431 Epithelial Cells Monolayer Infected with C. albicans in the Presence or Not of RBG

Vaginal epithelial cell infections were performed using 96-well plates (Costar 3595, Corning, New York, NY, USA). A-431 cells (5 × 10^5^ cells/mL, 200 μL per well) were seeded and then incubated overnight at 37 °C with 5% CO_2_. Before being infected, the epithelial cell monolayer was washed with Phosphate-Buffered Saline (PBS, Dutscher, Bernolsheim, France) and kept at room temperature. Then epithelial cells were infected with *C. albicans*, in the presence or not of RBG, by using a Multiplicity of Infection (MOI) of 1:1, namely 5 × 10^5^ CFU/mL.

### 2.5. Assessment of Vaginal Epithelial Cells Damage

Vaginal epithelial cell damage induced by treatment with RBGs at different dilutions (range dilutions 1:150–1:500) was evaluated after 24 h of incubation at 37 °C with 5% CO_2_. Cytotoxicity was quantified by analyzing lactate dehydrogenase (LDH) release in the culture medium employing a commercially available kit (Abcam, Cambridge, UK) and following the manufacturer’s instructions. LDH quantification was also performed for the assessment of *C. albicans*-induced vaginal cell damage in the presence or not of RBG or pRBG after 24 h of infection at 37 °C with 5% CO_2_. The percentage of damage was calculated as follows:(1)$$\%\;\mathrm{cell}\;\mathrm{damage}=\frac{\left(\mathrm{test}\;\mathrm{sample}-\mathrm{low}\;\mathrm{control}\right)}{\left(\mathrm{high}\;\mathrm{control}-\mathrm{low}\;\mathrm{control}\right)}\times 100$$
where low control is represented by uninfected cells (basal LDH release), and high control is given by uninfected cells lysed with 0.2% Triton X-100 after the incubation (maximum LDH release).

### 2.6. RBGs Impact on C. albicans Growth during Vaginal Epithelial Cell Infection

The effect of RBG on *C. albicans’* growth capacity was assessed by using the bioluminescent strain gLUC59 (BLI-Ca). The epithelial cell monolayer was infected with 200 μL of BLI-Ca (5 × 10^5^ CFU/mL) in the presence or not of RBG or pRBG. After 24 h of incubation at 37 °C with 5% CO_2_, coelenterazine (2 μM) was added to each well, and the bioluminescent signal was acquired with a Luminometer (Fluoroskan™ Microplate reader, Thermo Scientific, Waltham, MA, USA). Data were expressed as Relative Luminescence Units (RLU).

### 2.7. RBGs Direct Effect on Candida Growth and Metabolic Activity

RBG anti-*Candida* direct effect and its impact on fungal metabolic activity were analyzed after 24 h of incubation. One hundred microliters of *C. albicans* (5 × 10^5^ CFU/mL) in RPMI supplemented with MOPS and glucose were seeded in a 96-well plate in the presence or not of RBG or pRBG and incubated at 37 °C for 24 h. After incubation, fungal growth was quantified by optical density (OD) measurement at 540 nm wavelength by a plate reader (Sunrise Tecan, Männedorf, Switzerland). Subsequently, the medium of each well was removed, and 100 μL of 2,3-bis(2-methoxy-4-nitro-5-sulfophenyl)-2H-tetrazolium-5-carboxanilide (XTT) solution, supplemented with menadione (1 μM), were added. The plate was covered with tinfoil and incubated at 37 °C for 3 h. Then, 80 μL were collected from each well, transferred to wells of another microtiter plate and color intensity was quantified by measuring OD at 492 nm wavelength.

### 2.8. Effect of RBGs on C. albicans Adhesion

*C. albicans* adhesion was evaluated after 2 h of incubation at 37 °C in the presence or not of RBGs. Two hundred microliters of *C. albicans* in DMEM supplemented with 5% FBS were seeded in a 96-well plate, and after incubation, a crystal violet (CV) assay was performed. Specifically, wells were washed with PBS to remove non-adherent fungal cells, and 100 μL of 1% CV was added to each well. After 5 min, CV was removed, and two washes with PBS were carried out before adding 100 μL of 33% acetic acid to each well. The adhesion of *C. albicans* was quantified by evaluating the optical density (OD) at 570 nm wavelength using a spectrophotometer (Sunrise, Tecan, Männedorf, Switzerland).

### 2.9. Evaluation of RBGs Effect on C. albicans Hyphal Formation

RBG effect on *C. albicans* ability to form hyphae was assessed in YPD broth supplemented with 10% FBS. One milliliter containing 5 × 10^5^ CFU of *C. albicans,* in the presence or not of RBG or pRBG, was added to a 24-well plate (Greiner Bio-One, Kremsmünster, Austria) and then incubated at 37 °C for 4 h. After incubation, 100 μL of each sample were recovered and placed onto a glass microscope slide. Yeast cells and hyphal fragments were counted (at least 3 fields for each experimental condition) using an optical microscope (Nikon Eclipse 80i, Nikon Corporation, Tokyo, Japan). Hyphae were not stained, and they were counted in Brightfield. Data were expressed as % of hyphal fragments.

### 2.10. Quantification of IL-1β and IL-8 Production after C. albicans Infection and LPS Stimulation of Vaginal Epithelial Cells in the Presence or Not of RBGs

The production of IL-1β and IL-8 by A-431 cells monolayer was analyzed after 24 h of *C. albicans* infection (MOI 1:1) or LPS (10 μg/mL) and LPS+ Dexamethasone (DEX) (LPS: 10 μg/mL + DEX: 1 mM) stimulation in the presence or not of RBG or pRBG, by using specific commercially available sandwich ELISA kits (Boster Bio, Pleasanton, CA, USA for both IL-1β and IL-8), according to the Manufacturers’ instructions.

### 2.11. Statistical Analysis

The Shapiro-Wilk test was used to analyze data distribution within each experimental group. Data from all the experiments were normally distributed, and statistical analyses were performed by the one-way ANOVA test followed by Tukey’s multiple comparisons tests. All statistical analyses were carried out using GraphPad Prism 9 software. Values of * *p* < 0.05, ** *p* < 0.01 and **** *p* < 0.0001 were considered statistically significant.

## 3. Results

### 3.1. RBG Reduced C. albicans-Induced Vaginal Epithelial Cell Damage without Affecting Fungal Growth

First, we set up the experimental protocol to analyze the effect of RBG and pRBG on vaginal cells in vitro. We performed dose-dependent experiments to identify the highest RBG concentration tolerated by the vaginal cells, i.e., the ones incapable of inducing cell damage. RBG showed to be well tolerated by vaginal epithelial cells starting from dilution 1:300 after 24 h of contact time. Placebo RBG (pRBG) did not show any cell toxicity at any dilution tested. Therefore, dilution 1:300 (highlighted in light green in Figure 1) was chosen for both RBG and pRBG (Figure 1), and such dilution was employed in all the subsequent experiments. Of note, the pH of RBG and pRBG at dilutions from 1:150 to 1:300 was always around 7.4, resulting in it being superimposable to the pH of the medium alone.

Next, *C. albicans*-induced epithelial vaginal cell damage after 24 h of infection was assessed in the presence or absence of RBG or pRBG. Our results show that RBG, but not pRBG, was able to significantly reduce *C. albicans*-induced cell damage by an inhibition rate of about 23% (Figure 2A).

In parallel, fungal growth was assessed after 24 h of vaginal epithelial cells infected with BLI-*C. albicans* (BLI-Ca) in the presence or absence of RBG and pRBG. Results show that neither RBG nor pRBG were able to significantly impair *C. albicans* growth (Figure 2B) during epithelial cells infection, suggesting that the observed reduction in cell damage was not due to a direct antifungal effect.

Interestingly, by analyzing the direct antifungal activity of RBG and pRBG against *C. albicans* without cells monolayer, we observed a slight but significant reduction in *Candida* growth with RBG but not with pRBG (Figure 3A).

In contrast, by analyzing the metabolic activity of *C. albicans* under the same experimental conditions, no significant differences could be observed between *C. albicans* alone and RBG or pRBG-treated *C. albicans* (Figure 3B).

### 3.2. RBG Reduced C. albicans Adhesion and Hyphae Formation

To investigate the mechanism involved in the inhibition of *C. albicans*-induced cell damage by RBG, we evaluated its impact on fungal adhesion. RBG was shown to significantly reduce the number of adherent *C. albicans* cells after 2 h of incubation at 37 °C, with an inhibition rate of about 26%. The placebo formulation did not cause any inhibition (Figure 4A). As filamentation is a key virulence trait implicated in *C. albicans*-induced cell damage, as well as adhesion to biotic and abiotic substrates, we analyzed the capacity of RBG to modulate hyphae formation. Our results show that RBG significantly reduced *C. albicans* filamentation with an inhibition rate of about 31%, whereas pRBG did not cause any inhibitory effect (Figure 4B,C). Notably, by comparing pRBG with RBG, a significant difference could be observed for both adhesion and hyphal formation (Figure 4A–C).

### 3.3. RBG Effects on IL-1β and IL-8 Production by Vaginal Epithelial Cells

We next analyzed the capacity of RBG to modulate the epithelial immune response to *C. albicans* infection and to LPS stimulation. Specifically, we assessed IL-1β and IL-8 production after 24 h infection of vaginal epithelial cells with *C. albicans* in the presence or absence of RBG and pRBG. Our results show that *C. albicans* could induce the release of a significant (albeit modest) amount of IL-1β by vaginal epithelial cells and that such an amount was not modulated by either RBG or pRBG. In addition, the 24 h stimulation of vaginal epithelial cells with LPS induced the secretion of a small amount of IL-1β, not significantly different from the amount secreted by cells alone. No modulation of IL-1β secretion was observed after epithelial stimulation with LPS + DEX or LPS + RBGs; moreover, no modulation of IL-1β secretion was detected in the presence of RBG or pRBG alone (Figure 5A).

In contrast, *C. albicans* did not induce the secretion of IL-8 by vaginal epithelial cells after 24 h of infection under our experimental conditions. Accordingly, no effect could be observed by infecting vaginal epithelial cells with *C. albicans* + RBG or pRBG. However, LPS strongly stimulated IL-8 production by vaginal epithelial cells after 24 h. By adding the RBG and pRBG in combination with LPS, we observed a significant reduction in IL-8 with respect to the stimulation by LPS alone. As expected, the addition of DEX reduced the IL-8 levels in a highly significant manner. Interestingly, such reduction was comparable to the reduction induced by RBG (Figure 5B).

## 4. Discussion

In this work, we tested the antifungal and anti-inflammatory properties of the vaginal gel RBG, a commercially available formulation used as an adjuvant in the treatment of vaginitis and vaginosis. The manufacturer claims that this gel helps to restore the physiological pH of the vaginal environment and to promote the growth of a balanced microbiota [41]. De Seta and Larsen recently showed in an in vitro study that RBG gel has antimicrobial properties against several microorganisms relevant to vaginal infections, such as *Candida* spp. and *Gardnerella vaginalis*. They concluded that the RBG formulation contains different components, including some excipients (such as EDTA), that can play a role in counteracting microbial growth. Although they ascribed the antimicrobial effect mainly to the lactic acid, they hypothesized an additional mechanistic and possible synergistic interaction between active components (polydextrose, lactic acid, farnesol) that may occur when RBG encounters vaginal fluids [41]. We excluded an effect of lactic acid in our experimental conditions since the pH of both RBG and the placebo formulation is around 7.4 when diluted from 1:150 to 1:300, and therefore superimposable to the pH of the medium alone. Moreover, RBG contains prebiotic compounds (such as glycogen and polydextrose) [39,40] that can promote the growth of a balanced microbiota, thus contributing to counteracting potentially pathogenic microorganisms in the vaginal environment. Therefore, it is very likely that the main role of glycogen, polydextrose, and lactic acid must be exerted in the actual application.

In this work, we provided new data on RBG, studying its activity against *C. albicans* by an in vitro model that could mimic a physiological situation as closely as possible. Specifically, a monolayer of vaginal epithelial cells was infected with *C. albicans* in the presence or absence of the commercially available RBG (RBG) and a placebo formulation (pRBG). Our results clearly show that only RBG, but not pRBG, is able to reduce *Candida*-induced cell damage. It should be noted that RBG has been employed at a dilution of 1:300, i.e., the highest non-toxic dose for epithelial cells. This dilution is far below the doses tested by De Seta and Larsen to study the direct antimicrobial activity in vitro [41].

In parallel, during epithelial cell infection, *Candida* growth has not been affected by the addition of any of the two RBG formulations. Therefore, we hypothesize that the observed reduction in cell damage in our experimental model is not due to a direct antifungal effect exerted by RBG but rather to a reduction in the ability of *Candida* to express its virulence potential (for example, its capacity to adhere and/or to produce hyphae). There are two possible reasons that may explain this apparent discrepancy in our results with respect to De Seta and Larsen’s. First, as mentioned above, the dilution used in our protocol (1:300) is far below the doses tested by De Seta and Larsen. Second, the effect of the gel formulation on *Candida*, when it infects an epithelium monolayer, mirrors a situation more similar to the one occurring in vivo, when the fungus dwells within a biological environment where it can react more efficiently. It should be pointed out that, in line with the data obtained by De Seta and Larsen, by analyzing the direct antifungal activity of RBG against *C. albicans*, i.e., assessed without the vaginal cells monolayer, a slight, albeit significant, reduction in overall *Candida* growth has been detected after incubation with RBG, but not with pRBG. However, the results of the XTT test, used to evaluate the metabolic activity of the fungus, reveal that no significant differences could be observed between live *C. albicans* cells in the presence or absence of both RBG and pRBG.

The morphological transition from yeasts to hyphae represents a key step of *C. albicans* virulence because it increases deep tissue invasiveness and resistance to multiple environmental and chemical stresses [42,43]. Therefore, an effective strategy against *C. albicans* should include a reduction in its capacity to adhere and form hyphae rather than a mere growth reduction. Indeed, *C. albicans* could be part of the vaginal microbiota of healthy women, and in its yeast form, it is considered a harmless commensal [7].

In line with this idea, our results show that RBG significantly reduces the capacity of *C. albicans* to adhere; in addition, RBG impairs *C. albicans*’ capacity to form hyphae. We tend to ascribe these effects essentially to farnesol since, as mentioned above, the role played by the other three active components (lactic acid, polydextrose and glycogen) would become evident during the treatment. Farnesol is a quorum-sensing (QS) molecule that, together with tyrosol, is involved in biofilm formation through the modulation of several virulence factors of the fungus, including the dimorphic transition. In particular, while tyrosol stimulates hyphae production, therefore exerting a pivotal role in biofilm formation, farnesol counteracts yeast-to-hyphae transition (one of the effects of RBG) and inhibits biofilm formation [44]. In addition, glycogen and polydextrose are two active components of RBG, with prebiotic activity. Both may favor the proliferation of “beneficial” bacteria, which are part of the healthy vaginal microbiota. Indeed, a glycogen-rich vaginal milieu has been demonstrated to favor the proliferation of lactobacilli, which in turn increases the production of lactic acid and decreases the pH [45]. Therefore, the role of glycogen in the RBG formulation must be indirect by exerting a positive effect on the vaginal microbiota. It is well-known that a lactobacilli-dominated microbiota provides an effective first line of defense against invading pathogens, including *Candida* [46], through several mechanisms: competition for adhesion sites on the host cells, competition for nutrients, and release of antimicrobial compounds such as bacteriocins, hydrogen peroxide and organic acids [47]. Similarly, polydextrose should also be considered a prebiotic component, capable of playing a role in vivo stimulating the growth of the “good” bacteria.

The role of the inflammatory response on vaginal infection onset is well documented [7,48]. Concerning vulvovaginal candidiasis (VVC), although fungal virulence factors are important for triggering the disease, its spread is largely mediated by the host immune system. In particular, *C. albicans’* overgrowth, yeast-to-hyphae transition and candidalysin production stimulate the epithelial cells to secrete antimicrobial peptides and proinflammatory cytokines and chemokines, such as IL-1β and IL-8. This leads to the massive recruitment of non-protective neutrophils that, in turn, generate a hyper-inflammatory environment mainly responsible for the VVC symptoms [7]. Similarly, numerous studies have demonstrated the occurrence of high levels of cytokines and chemokines, such as IL-1β, TNF-α, IL-6 and IL-8, in vaginal fluids of women with bacterial vaginosis (BV) [48].

Through our in vitro model, we analyzed the capacity of RBG to modulate cytokines and chemokines production by epithelial cells during *C. albicans* infection or after treatment with LPS. Our results show that both RBG and pRBG display some degree of activity in reducing LPS-induced IL-8 production. Notably, although both RBG and its placebo significantly reduce IL-8 secretion by LPS-stimulated epithelial cells, the reduction observed after RBG treatment is comparable to that observed after DEX treatment. EDTA (an excipient contained in both RBG formulations) has been reported to enhance the activity of host defense factors, including lysozyme and lactoferrin [49], both of which are also components of vaginal secretions [50,51]. Therefore, our hypothesis is that by potentiating the epithelial barrier, both RBG and pRBG are able to reduce LPS-induced IL-8 secretion.

## 5. Conclusions

These results expand the data by De Seta and Larsen [41] because RBG has the capacity to impair *Candida* virulence by counteracting *C. albicans*-induced vaginal epithelial cell damage. Such effect seems to be achieved through the impairment of specific virulence factors of the fungus, i.e., the reduction in its capacity to adhere and to undergo the yeasts-to-hyphae morphological transition, which are two key factors in fungal pathogenesis. Interestingly, the gel formulations seem not to impair either fungal growth or its metabolic activity. Consequently, the reduced epithelial cell damage exerted by RBG, but not by pRBG, is not achieved through a reduction in the fungal burden. It is our hypothesis that, among the active compounds, the main role may be played by farnesol, which is a well-known QS *Candida* molecule whose role in promoting yeast morphology has been described [44]. Accordingly, the literature data report that farnesol does not impair *Candida* growth [22]. The effects of RBG in our experimental system are summarized in Figure 6. This mechanism of action highlights the impact of farnesol without including the role of lactic acid, polydextrose and glycogen. As mentioned above, the effects of such three active components are relevant mainly in vivo.

In addition, some of the excipients contained in both RBG and pRBG (such as EDTA) may help in reducing the IL-8 secretion by LPS-stimulated epithelial cells, and such effect is superimposable to the same inhibition obtained after treatment with DEX. The resulting IL-8 secretion inhibition may be responsible (in combination with other mechanisms) for the recovery of a low inflammatory vaginal environment, which in turn is the starting point for the establishment of vaginal eubiosis. In addition, such effect may be strengthened by glycogen, polydextrose and lactic acid contained in RBG that, by helping the proliferation of lactobacilli, provide another means for the vaginal environment to prevent invasion of potential pathogens like *Candida*.

## Figures and Tables

**Figure 1 microorganisms-11-01551-f001:**
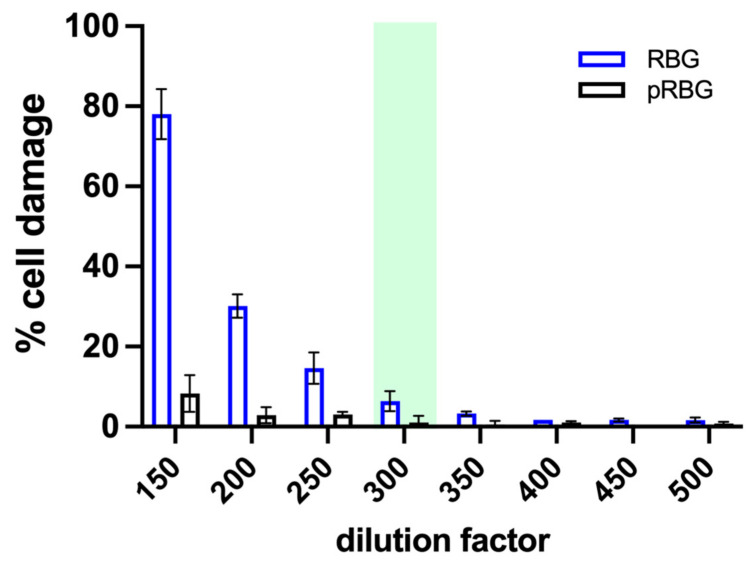
Effect of RBG on vaginal epithelial cells. Cell damage was evaluated by the quantification of lactate dehydrogenase (LDH) release in the culture medium after 24 h of treatment with RBG and pRBG. The graph reports the mean percentage of damage ± SD from triplicate samples of at least 3 different experiments (range dilutions 1:250–1:350) and from triplicate samples of at least 2 different experiments (dilution 1:150; 1:200 and from 1:400).

**Figure 2 microorganisms-11-01551-f002:**
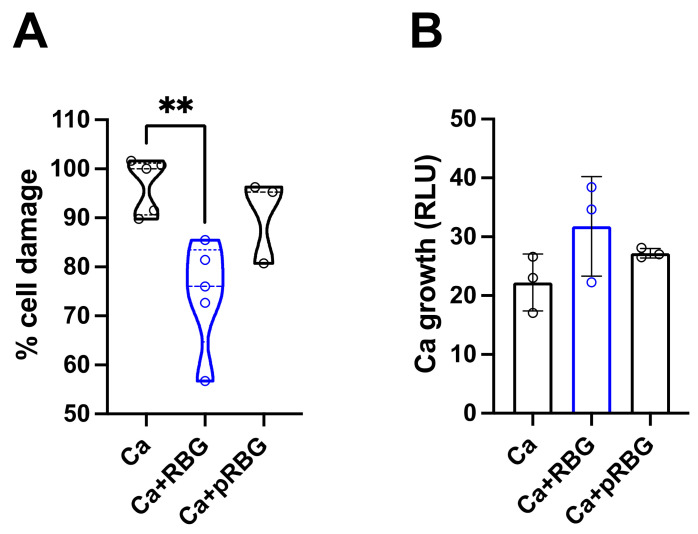
Effect of RBG on *C. albicans*-induced cell damage and on fungal growth during vaginal cell infection: (**A**) Epithelial cell damage was evaluated by the quantification of lactate dehydrogenase (LDH) release in the culture supernatants after 24 h of *C. albicans* infection in the presence or absence of RBG and pRBG. Data reported in the violin plot are from triplicate samples of at least 3 different experiments. ** *p* < 0.01 (**B**) *C. albicans* growth was analyzed by the determination of Relative Luminescence Units (RLU) emitted from live fungal cells after 24 h of infection of vaginal epithelial cells in the presence or absence of RBG and pRBG. The graph reports the mean RLU ± SD from triplicate samples of 3 different experiments.

**Figure 3 microorganisms-11-01551-f003:**
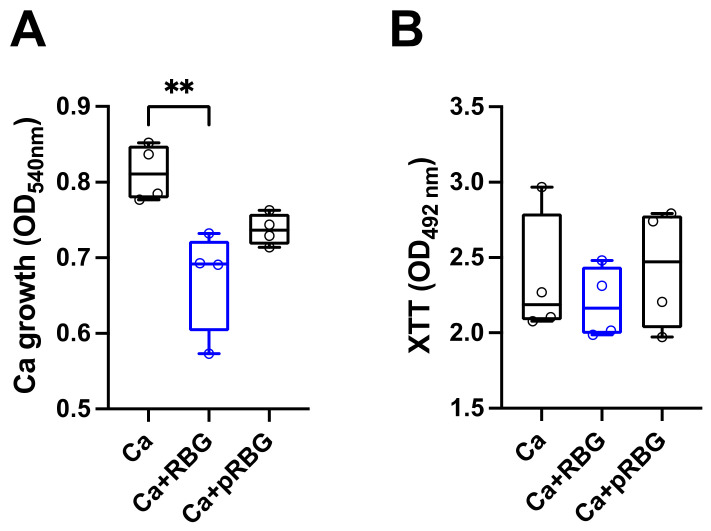
RBG effects on *C. albicans* growth and metabolic activity: (**A**) The anti-*Candida* effect of RBG was analyzed by culturing *C. albicans* yeast cells in the presence or absence of RBG and pRBG for 24 h. After incubation, absorbance (OD_540_) was spectrophotometrically quantified. Data in the box plots are from triplicate samples of 4 different experiments. ** *p* < 0.01 (**B**) The metabolic activity of *C. albicans* after 24 h of incubation with RBG and pRBG was analyzed by XTT assay. Data in the box plots are from triplicate samples of 4 different experiments.

**Figure 4 microorganisms-11-01551-f004:**
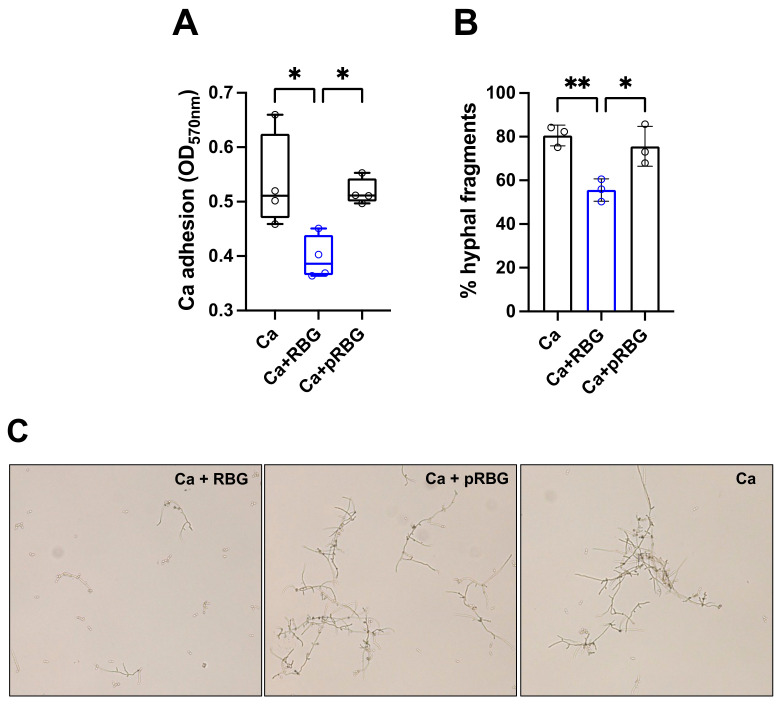
RBG effect on *C. albicans* adhesion and hyphae formation: (**A**) Effect of RBG on *C. albicans* adhesion to abiotic surface after 2 h of incubation at 37 °C in DMEM supplemented with 5% FBS. Data in the box plots are from triplicate samples from 4 different experiments. * *p* < 0.05 (**B**) RBG effect on *C. albicans* hyphal formation after 4 h of incubation at 37 °C in YPD liquid medium + 10% FBS. Data are expressed as the mean percentage of hyphal fragments ± SD from at least 3 analyzed fields for each condition from 3 different experiments. ** *p* < 0.01, * *p* < 0.05 (**C**) Representative fields of RBG or pRBG effect on *C. albicans* hyphal formation as detailed above. Magnification 20×.

**Figure 5 microorganisms-11-01551-f005:**
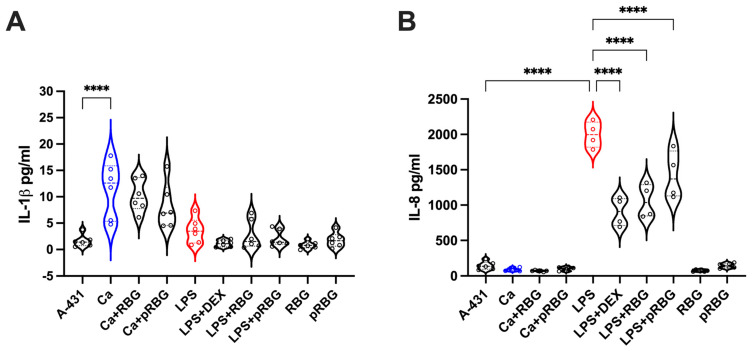
Interleukins production: IL-1β (**A**) and IL-8 (**B**) production by vaginal epithelial cells after 24 h of *C. albicans* (Ca) infection or LPS treatment at 37 °C plus 5% CO_2_ in the presence or absence of RBG, pRBG and dexamethasone (DEX). Violin plot shows the interleukins amounts released in the culture medium. Data are expressed as pg/mL of duplicate samples from at least 2 different experiments. **** *p* < 0.0001.

**Figure 6 microorganisms-11-01551-f006:**
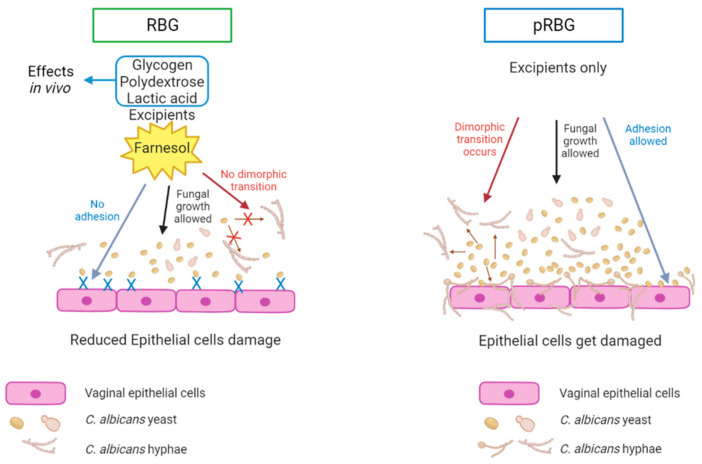
Proposed model of RBG effects on *C. albicans*. Farnesol included in RBG counteracts yeast-to-hyphae transition and *C. albicans* adhesion without affecting *Candida* growth. This leads to a reduced capacity of *C. albicans* to damage vaginal epithelial cells. This scheme highlights the impact of farnesol without including the role of lactic acid, polydextrose and glycogen. However, the effects of such 3 components must be relevant in the actual application (**left** panel). pRBG is not able to modulate *C. albicans* morphological transition, adhesion, and growth. Hence, *C. albicans* can induce cell damage unchallenged (**right** panel).

**Table 1 microorganisms-11-01551-t001:** Composition of RBG and pRBG. Components identified as active compounds are written in bold blue.

RBG	pRBG
Water	Water
Disodium EDTA 0.2%	Disodium EDTA 0.2%
Xanthan gum	Xanthan gum
Sodium hyaluronate	Sodium hyaluronate
Propylene glycol	Propylene glycol
Decylene glycol	Decylene glycol
Hydroxyacetophenone	Hydroxyacetophenone
Hydrogenated castor oil	Hydrogenated castor oil
Tocopherol acetate	Tocopherol acetate
PEG-40	PEG-40
** Polydextrose **	
** Lactic acid **	
** Farnesol **	
** Glycogen **	

## Data Availability

Not applicable.

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
