# Peer review of "Anti-Candida and Anti-Inflammatory Properties of a Vaginal Gel Formulation: Novel Data Concerning Vaginal Infection and Dysbiosis"

_microorganisms, 2023, doi:10.3390/microorganisms11061551_

Round 1

Reviewer 1 Report

Dear authors,

qualified investigations on the effects of pre- and probiotics  in the vaginal environment are sparse. I appreciate therefore your study with this gel containing glycogen, lactic acid, farnesol and poly dextrose on Candida albicans and its interactions with the vaginal epithelium. I did not find any reason for a correction in the methods, results or the conclusions.

Author Response

Response: We would like to thank you the Reviewer for his/her kind comments

Reviewer 2 Report

The manuscript shows the antifungal and anti-inflammatory properties of RBG vaginal gel. The manuscript is well developed based on justifications and hypotheses about the ability of RBG to reduce C. albicans adhesion, its antimicrobial effect, and possible mechanisms involved in inhibiting Candida-induced cell damage. With this, minor considerations need to be made for better scientific rigor of the article:

Line 147: The authors did not indicate what were the dilution ranges needed to assess the effect of RBGs on vaginal epithelial cells;

Line 190: For hyphae formation your microscopic analysis, were the samples treated and stained?

Line 275: Was the quantification of the production of pro-inflammatory cytokines IL-6, IL-7, and IL-13 performed by the authors?

Author Response

The manuscript shows the antifungal and anti-inflammatory properties of RBG vaginal gel. The manuscript is well developed based on justifications and hypotheses about the ability of RBG to reduce C. albicans adhesion, its antimicrobial effect, and possible mechanisms involved in inhibiting Candida-induced cell damage. With this, minor considerations need to be made for better scientific rigor of the article:

Line 147: The authors did not indicate what were the dilution ranges needed to assess the effect of RBGs on vaginal epithelial cells;

Response: Thank you. Amended (page 4, line 148).

Line 190: For hyphae formation your microscopic analysis, were the samples treated and stained?

Response: Thank you for the observation. Hyphae were not stained, and they were counted in brightfield. This information is now included in the revised manuscript (page 5, line 192-193).

Line 275: Was the quantification of the production of pro-inflammatory cytokines IL-6, IL-7, and IL-13 performed by the authors?

Response: We thank the Reviewer for this observation. We analyzed IL-6 but we did not observe any significant modulation of this cytokine in response to Candida, LPS or RBG. Concerning IL-7 and IL-13 we will included the determination of these cytokines in the future experiments.